# Targeted Anticancer Agent with Original Mode of Action Prepared by Supramolecular Assembly of Antibody Oligonucleotide Conjugates and Cationic Nanoparticles

**DOI:** 10.3390/pharmaceutics15061643

**Published:** 2023-06-02

**Authors:** Victor Lehot, Patrick Neuberg, Manon Ripoll, François Daubeuf, Stéphane Erb, Igor Dovgan, Sylvain Ursuegui, Sarah Cianférani, Antoine Kichler, Guilhem Chaubet, Alain Wagner

**Affiliations:** 1Bio-Functional Chemistry (UMR 7199), Institut du Médicament de Strasbourg, University of Strasbourg, 74 Route du Rhin, 67400 Illkirch-Graffenstaden, France; 2UAR3286, Plate-Forme de Chimie Biologique Intégrative de Strasbourg, ESBS, CNRS-Strasbourg University, 67400 Illkirch-Graffenstaden, France; 3Laboratoire de Spectrométrie de Masse BioOrganique (LSMBO), Institut du Médicament de Strasbourg, Université de Strasbourg, CNRS, IPHC UMR 7178, 67000 Strasbourg, France

**Keywords:** antibody conjugate, nanoparticle, supramolecular conjugation, anticancer, targeting

## Abstract

Despite their clinical success, Antibody-Drug Conjugates (ADCs) are still limited to the delivery of a handful of cytotoxic small-molecule payloads. Adaptation of this successful format to the delivery of alternative types of cytotoxic payloads is of high interest in the search for novel anticancer treatments. Herein, we considered that the inherent toxicity of cationic nanoparticles (cNP), which limits their use as oligonucleotide delivery systems, could be turned into an opportunity to access a new family of toxic payloads. We complexed anti-HER2 antibody-oligonucleotide conjugates (AOC) with cytotoxic cationic polydiacetylenic micelles to obtain Antibody-Toxic-Nanoparticles Conjugates (ATNPs) and studied their physicochemical properties, as well as their bioactivity in both in vitro and in vivo HER2 models. After optimising their AOC/cNP ratio, the small (73 nm) HER2-targeting ATNPs were found to selectively kill antigen-positive SKBR-2 cells over antigen-negative MDA-MB-231 cells in serum-containing medium. Further in vivo anti-cancer activity was demonstrated in an SKBR-3 tumour xenograft model in BALB/c mice in which stable 60% tumour regression could be observed just after two injections of 45 pmol of ATNP. These results open interesting prospects in the use of such cationic nanoparticles as payloads for ADC-like strategies.

## 1. Introduction

Antibody-drug conjugates (ADCs) associate the exquisite targeting abilities of antibodies with the cytotoxicity of various small-molecule payloads through covalent bioconjugation, resulting in antigen-specific cell killing [1]. With the exception of one (Moxetumomab pasudotox, whose payload is a fused PE38 toxin altering protein synthesis by inhibition of the elongation factor 2) [2], the payloads of all currently FDA-approved ADCs are small cytotoxic drugs targeting either tubulin, DNA or topoisomerase [3,4]. Despite the clinical success of ADCs, this narrow set of payloads limits their widespread use. [5] For example, tubulin inhibitors’ toxicity is limited by their selectivity for proliferating cells, while DNA-targeting payloads have raised safety challenges due to their non-specific uptake in sensitive normal tissues [3]. Thus, there is a need for innovative types of active payloads for ADCs, with several avenues being explored such as conjugations of antibodies to therapeutic oligonucleotides or drug-loaded nanoparticles [6,7].

In parallel, the therapeutic promises of gene therapy have driven the emergence of a vast panel of intracellular delivery systems [8,9,10,11,12]. Among the many strategies designed, electrostatic complexation of anionic oligonucleotides with cationic nanoparticles (cNP) [11,13,14,15] is one of the most fruitful. Although promising as vectors, various types of cNP have been shown to cause important cytotoxicity [16,17,18,19,20]. Their positive charges promote interaction with the negatively charged phospholipids in cellular membranes [21,22]. These interactions disturb the integrity of the cytosolic membrane, but also that of membranes of intracellular compartments such as mitochondria and lysosomes. This can cause the formation of holes, leading to various events such as influx of Ca^2+^ and intracellular release of cytochrome C (from mitochondria) and hydrolases (from lysosomes), leading to cell death [19,20,23]. In the context of anticancer drug delivery, such cytotoxicity can be seen as a convenient additional property [24] or even be deliberately used as a primary mode of action for cell killing [25].

We hypothesised that the toxicity of cNP, considered to be a limitation in the field of gene therapy, could address the need for novel ADC payloads. Association of nanoparticles as payload with targeting agents in the field of nanomedicine is mostly achieved via covalent conjugation [7,26,27]. Alternatively, more flexible strategies based on supramolecular assembly have proven efficient in controlling and optimising molecular architectures of siRNA targeted delivery vehicles [28]. However, Ab and cNP do not spontaneously assemble via supramolecular interactions. In order to trigger such programmed association, we exploited the innate affinity of cNP for oligonucleotides by modifying Abs with ssDNA. Interestingly, analogous strategies based on electrostatic interactions have allowed for the assembly of charged nanoparticles with either active targeting agents [29,30] or protein-oligonucleotide cargo [31]. Antibodies were then covalently conjugated with small oligonucleotides serving as molecular glue enabling them to stick to cNP and form a novel type of supramolecular anticancer Antibody-Toxic Nanoparticles (ATNP; Figure 1).

An ideal cNP payload should bear a strong affinity for oligonucleotides, in order to stabilise the non-covalent linkage with AOC, and upon acidification in lysosomes, maintain its ability to interact with cellular membranes and intracellular compartments and ultimately cause cell death. Cationic photopolymerised polydiacetylenic (PDA) micelles are a promising class of cNP with modular surface chemistry, high loading capacity and easy assembly [32]. Moreover, the photopolymerisation process was shown to increase PDA micelles’ stability [33,34]. For the particular purpose of this paper, we designed a tyrosine-histidine dipeptide cationic polar head group that provides PDA with high affinity for both oligonucleotides and cellular membranes. [35] Indeed, histidine modification of cNPs has been shown to increase transfection efficiency, endosomal escape abilities (which allow further interactions with intracellular compartments) [34,36], and serum tolerance [37]. On the other hand, tyrosine modification yields fusogenic properties through hydrophobic interactions with membranes’ phospholipids, which enhances cellular uptake and endosomal escape and can lead to increased cytotoxicity [38,39].

The well-established drug trastuzumab (T), which recognises the human epidermal growth factor receptor 2 (HER2) overexpressed in certain breast cancer cells, was selected for this study [40]. As a control to validate that the cell-killing was indeed mediated by antigen recognition, rituximab (R), an anti CD20 Antibody, an antigen present on B-cells only, was used [41]. For the oligonucleotide moiety of our conjugates, we selected single stranded DNA (ssDNA) whose sequence was designed to prevent any folding (using IDT’s UNAFold tool), maximising its availability in the interaction with the cationic particle. 

## 2. Materials and Methods

### 2.1. Materials

^1^H and ^13^C NMR spectra were recorded at 23 °C on Bruker Avance III–400 MHz/500 MHz spectrometers. Recorded shifts are reported in parts per million (δ) and were calibrated using residual non-deuterated solvent. Data are represented as follows: chemical shift, multiplicity (s = singlet, d = doublet, t = triplet, m = multiplet), coupling constant (J, Hz), integration in the case of 1H NMR data. High-resolution mass spectra (HRMS) were obtained using an Agilent Q-TOF 6520.

Native mass spectrometry (native MS) analyses of intact proteins and conjugates were performed using a Waters LCT mass spectrometer coupled to an automated chip-based nanoESI infusion source (Triversa Nanomate, Advion, Ithaca, NY, USA), both operating in positive ion mode. Electrospray ionisation was conducted at a capillary voltage of 1.75 kV and nitrogen nanoflow of 0.75 psi. Samples were directly infused after manual desalting at a concentration of 10 μM. 

All reagents were obtained from commercial sources and used without prior purification. Amino-modified (5AmMC12) oligonucleotides were purchased from IDT. Protein-oligonucleotide conjugates were purified by gel filtration using ÄKTA Pure System (isocratic elution with DPBS 1x, pH 7.5, 0.5 mL/min, column: Superdex 200 Increase 10/300 GL). The oligonucleotide species were purified using a Shimadzu HPLC system (pumps: LC 20-AD, detector: SPD 20-A, autosampler: SIL 20-A) using a XTerra MS C18 5 μM 4.6 × 150 mm column (Waters), with a flow rate of 1 mL/min (Mobile phase: A triethylammonium acetate 50 mM in water, B triethylammonium acetate 50 mM in acetonitrile).

Hepes buffered glucose (HBG): 20 mM HEPES, glucose 5%, pH 7.5.

### 2.2. Methods

#### 2.2.1. Micelle Synthesis

Tyrosine-PDA monomer (419 mg, 566 µmole, 1 equiv., synthesised as previously described [42]) was placed in DCM (30 mL) along with activated molecular sieves. Then were added, in order, Boc-His(Boc)-OH (365 mg, 679 µmole, 1.2 equiv.), HBTU (322 mg, 849 µmole, 1.5 equiv.), HOBt (115 mg, 849 µmole, 1.5 equiv.) and DIPEA (281 µL, 1.7 mmole, 3 equiv.) and the reaction was allowed to proceed over 3 days. The crude mixture was then concentrated under vacuum and purified using flash chromatography (DCM/MeOH 0% to 7%). Fractions containing mono- and di-Boc-protected coupling products were isolated and concentrated under vacuum. This mixture was dissolved in DCM (14 mL), supplemented with TFA (2 mL), and stirred at 0 °C for 3 h. The desired Tyrosine-Histidine-PDA monomer was purified via flash chromatography (DCM/MeOH/NH_3_ 95/4.5/0.5 to 88/10.8/1.2) and obtained in 40% yield (198 mg). ^1^H NMR (400 MHz, MeOD) δ: 7.64 (s, 1H), 7.03 (d, *J* = 8.6 Hz, 2H), 6.88 (s, 1H), 6.70 (d, *J* = 8.6 Hz, 2H), 4.47 (t, *J* = 7.3 Hz, 1H), 3.73–3.68 (m, 1H), 3.65–3.47 (m, 10H), 3.39–3.14 (m, 6H), 3.03–2.91 (m, 2H), 2.90–2.82 (m, 2H), 2.24 (t, *J* = 6.9 Hz, 4H), 2.17 (t, *J* = 7.5 Hz, 2H), 1.79–1.70 (m, 2H), 1.70–1.63 (m, 2H), 1.63–1.56 (m, 2H), 1.54–1.46 (m, 4H), 1.43–1.26 (m, 26H), 0.90 (t, *J* = 7.0 Hz, 3H); ^13^C NMR (100 MHz, MeOD) δ: 174.8, 172.7, 171.8, 156.0, 135.3, 130.0, 127.3, 116.9, 114.9, 76.5 (2C), 70.1, 69.8 (2C), 68.5, 68.2, 65.0, 55.1, 54.1, 37.0, 36.4, 36.3, 35.8, 31.7, 30.9, 29.3 (2C), 29.2, 29.1, 28.8 (3C), 28.7, 28.6, 28.4 (2C), 28.1, 25.6, 22.3, 18.3, 13.0; HRMS (ESI^+^) calcd for C_50_H_80_N_6_O_7_Na [M + Na]^+^ 899.5986; found 899.5969.



#### 2.2.2. Micelle Formulation and Photopolymerisation

Photopolymerised micelles were prepared according to previous reports. Briefly, 5 mg of Tyrosine-Histidine-PDA monomer was placed in 800 µL of ethanol and 200 µL of an HCl 0.1 M aqueous solution, and sonicated until total solubilisation (ca. 5 min, 80 W, 25 °C). The solution was then evaporated under reduced pressure, leading to the formation of a film. The obtained film was dissolved in 1 mL of deionised water and sonicated for 30 min (80 W, 25 °C), yielding self-assembled micelles. The obtained micelle solution was UV-irradiated at 254 nm and 48 W in 1 mL quartz cuvettes using a Cross-Linker Bio-Link 254 for 4 h. Finally, the photopolymerised micelles were purified via dialysis using 2000 MWCO dialysis cassettes from 7/3 EtOH/H_2_O to H_2_O.

#### 2.2.3. Conjugate Synthesis

Oligonucleotide Functionalisation:

BCN-PEG_6_-PFP was synthesised as previously described [43]. In a 2 mL Eppendorf tube, 5′-amino-modified oligonucleotide (1 equiv., 50 µL, 1 mM in water) was combined with BCN-PEG_6_-PFP (20 equiv., 50 µL, 20 mM in DMSO) and NaHCO_3_ (100 equiv., 5 µL, 1 M in water). The mixture was incubated at 25 °C overnight. The mixture was then diluted with water to a final volume of 300 µL and combined with acetone (900 µL) and LiClO_4_ (20 µL, 3 M in water) in order to precipitate the oligonucleotide species. The sample was then centrifuged (15,000× *g*, 8 min) and the supernatant was discarded. The precipitate was dissolved with water (300 µL) to repeat the precipitation and centrifugation procedure a second time.

Oligonucleotide Purification:

The previously obtained precipitate was dissolved with water (100 µL) and purified via HPLC (detection at 260 nm, mobile phase gradient A/B 9:1 to 6:4 in 30 min). After lyophilisation, the oligonucleotide conjugate was dissolved in DPBS (1×, pH 7.4) and analysed by absorption spectrophotometry (measured at 260 nm using a Nanodrop) to calculate the solution’s concentration using the Beer–Lambert law.

Antibody Azido-Functionalisation

4-azidobenzoyl fluoride (ABF) was synthesised as previously described [43]. ABF (3 equiv., 10 mM in DMSO) was added to a solution of antibody (1 equiv., 5 mg/mL, 100 µL in DPBS 1×, pH 7.4) and the reaction mixture was incubated at 25 °C for 30 min. The excess reagent was then removed by gel filtration chromatography using Bio-spin P-30 Columns (Bio-Rad, Hercules, CA, USA) pre-equilibrated with DPBS (1×, pH 7.4) to give a solution of azido-antibody, which was used in the following step.

Antibody-Oligonucleotide Conjugate Synthesis

To a solution of the azido-antibody conjugate (1 equiv., 5.0 mg/mL, in 100 µL DPBS 1×, pH 7.4), 10 µL of DPBS 10x was added, to prevent the precipitation of the antibody-oligonucleotide conjugate which would be formed. Then, the previously obtained 5′-BCN-modified oligonucleotide (3 equiv., 0.5 to 1 mM in DPBS 1×) was added to the mixture, which was incubated for 24 h at 25 °C. The conjugates were purified via size exclusion chromatography using AKTA Pure System (isocratic elution with DPBS (1×, pH 7.4), 0.5 mL/min) to yield the antibody-oligonucleotide conjugates.

Fluorescein Labelling

5′-fluorescein labelled (56-FAM) ssDNA oligonucleotide was purchased from IDT, and used without further purification.

Trastuzumab-ssDNA conjugate was concentrated to 2 mg/mL using micro-concentrators (Vivaspin, 500 µL, 50 cutoff, Sartorius, Gottingen, Germany), and combined with 30 equiv. of FITC (10 mM in DMSO). The mixture was then incubated at 25 °C overnight. The excess of FITC was then removed via gel filtration using Bio-spin P-30 Columns (Bio-Rad, Hercules, CA, USA) pre-equilibrated with DPBS 1× (pH 7.5) to yield a solution of FITC-labeled conjugate with a degree of labelling of 5.7 (determined using the protocol below).

#### 2.2.4. Conjugate Characterisation

Sample Preparation for Native-MS Analysis

Prior to native MS analyses, azido-modified antibodies were deglycosylated by incubation with 0.4 unit of Remove-iT^®^ Endo S per µg of antibody, at 37 °C for 2 h. The deglycosylated conjugates were then desalted against 150 mM NH_4_Ac buffered at pH 7.4, using 10 cycles of concentration/dilution via micro-concentrators (Vivaspin, 500 µL, 50 cutoff, Sartorius, Gottingen, Germany). Antibody concentration was then determined via UV absorbance using a NanoDrop spectrophotometer (Thermo Fisher Scientific, Illkirch, France).

Antibody-Oligonucleotide Conjugates Concentration Determination

The concentration of a protein in a given solution can usually be determined by measuring its absorption at 280 nm and using the Beer–Lambert law. ON’s absorbance at 280 nm makes it impossible to determine the concentration of antibody-oligonucleotide conjugates through absorption spectrophotometry measurement.

Antibody-oligonucleotide conjugates’ concentration was therefore determined using a Pierce BCA protein assay kit (ThermoFisher ref 23225), following the manufacturer’s protocol. This method allows for quantification of the protein moiety’s concentration, regardless of the presence of conjugated ONs. Concentrations were used to calculate the yields of the conjugates.

Antibody-Oligonucleotide Conjugates DoC Distribution Determination via SDS PAGE and Native Mass Spectrometry (MS)

SDS-PAGE was performed on 4–20% Mini-PROTEAN TGX Gel (Bio-Rad ref 4561094) following the manufacturer’s procedure. Antibodies or antibody-oligonucleotide conjugates (8 µL, 0.2 mg/mL in DPBS 1×) were added to 3 µL of non-reducing Laemmli SDS sample buffer (Alfa Aesar), and heated at 95 °C for 5 min. The resulting solutions were deposited, and the gel was run at constant voltage (200 V) for 35 min using TRIS 0.25 M–Glycine 1.92 M–SDS 1% as a running buffer. Coomassie Blue staining was performed using InstantBlue solution, prior to visualisation on a GeneGenius bio-imaging system (Syngene, Frederick, MD, USA). 

The lines’ intensities were determined using Image Studio Lite 5.2 software (LI-COR Biosciences, Lincoln, NE, USA), and the DoC of each conjugate was calculated using the following formula (Equation (1)):(1)$$\mathrm{DoC}=\frac{\sum_{k}k\;\times \;I({\mathrm{DoC}}_{k})}{\sum_{k}I({\mathrm{DoC}}_{k})}$$
where I(DoC_k_) is the line’s intensity of the conjugate with k conjugated oligonucleotides per antibody. 

Additionally, we analysed the deglycosylated azido-modified trastuzumab intermediate using native mass spectrometry. As observed in a previous work [43,44], the mean DoC values obtained by the integration of theSDS-PAGE gel bands of the Ab-ssDNA conjugates (2.9; see Figure 2(Aa) and Appendix A) and the native MS of the azido-modified intermediates (2.8; see Figure 2(Ab), Appendix A) were closely correlated.

#### 2.2.5. Complex Characterisation

Gel Mobility Shift Assay

To perform gel mobility shift assays we prepared 1.3% agarose gel by dissolving 400 mg of agarose in 30 mL of tris-acetate 40 mM, pH 7.5. The gels were run at 80 V for 1 h. Fluorescence of the fluorescein-labeled conjugates or oligonucleotides was visualised using GE Healthcare LAS 4000. The quantities involved for each gel are provided in Appendix A, and images of the gels in Figure 3C, Appendix A.

Size and ζ Potential Measurement

Dynamic light scattering (DLS) experiments were performed using a Zetasizer Nano-ZS (Malvern Instruments) and a low-volume quartz cuvette (ZEN2112 QS 3.00 mm). Analysis of the results was carried out using Malvern Zetasizer Software v8.01. Size and zeta potential spectra are provided in Figure 2(Ba,Bb), respectively. Size and zeta potential values are plotted in Figure 3B.

The hydrodynamic diameters of the micelles and the ATNP complexes were measured in water with the following parameters: Temperature: 25 °C, Index of refraction of material: 1.43, Index of refraction of pure water: 1.33; Viscosity of water used: 0.8872 cP; Sampling time: 55 s; Number of runs: 3

The ζ potential measurements were carried out using the same instrument and samples, using a disposable folded capillary zeta cell (1 mL, Malvern Panalytical DTS1070) and an effective voltage of 70 V with the following parameters: Temperature: 25 °C, Index of refraction of material: 1.43, Index of refraction of pure water: 1.33

Viscosity of water (and sample) used: 0.8872 cP, Dielectric constant: 78.5, Model: Smoluchowski, Number of runs: 3. The size and zeta potential measured are reported in Appendix A.

Monomer per Particle Estimation

The Van Der Waals volume of the monomeric Y was estimated using the formula described by Zhao et al. [45]:(2)$$V_{\mathrm{monomer}}=7.24\times N_{H}+20.58\times N_{C}+15.6\times N_{N}+14.71\times N_{O}-5.92\times \mathrm{NB}-14.7\mathrm{RA}-3.8\mathrm{RNR}$$
where V_monomer_ is the volume of the monomer, N_H_, N_C_, N_N_ and N_O_ are the numbers of hydrogen, carbon, nitrogen and oxygen atoms in the molecule, respectively, and NB, RA and RNR are the numbers of bonds, aromatic rings and non-aromatic rings in the molecule, respectively.

Using this volume and the diameters obtained from DLS measurements, we calculated the volume of the micelles using Equation (3), and finally, the number of monomers per particle using Equation (4).
(3)$$V_{\mathrm{micelle}}=\frac{4}{3}\times \pi \times {\left(\frac{d}{2}\right)}^{3}$$
where V_micelle_ is the volume of the micelle and d is the diameter measured by DLS.
(4)$$\mathrm{Monomer}\;\mathrm{per}\;\mathrm{particle}=\frac{V_{\mathrm{micelle}}}{V_{\mathrm{monomer}}}$$
where V_monomer_ is the volume of the monomer, and V_micelle_ is the volume of the micelle.

The quantities involved in the preparation of the various complexes formed throughout this study, as well as the calculated antibody-oligonucleotide conjugate/nanoparticle ratios, are indicated in Appendix A.

### 2.3. In Vitro Experiments

#### 2.3.1. Cell Culture

Human breast adenocarcinoma cells SK-BR-3 (ATCC HTB-30) and MDA-MB-231 (ATCC HTB-26)) were grown in Dulbecco’s Modified Eagle’s Medium (DMEM) containing 4.5 g/L glucose (Sigma, St Louis, MO, USA). The medium was supplemented with 10% fetal bovine serum (Perbio, Brebieres, France), 2 mM L-Glutamine, 100 U/mL Penicillin and 100 µg/mL Streptomycin (Sigma). Cells were maintained in a 5% CO2 humidified atmosphere at 37 °C.

#### 2.3.2. In Vitro Cytotoxicity Assay

The day before the experiment, SKBR3 and MDA-MB-231 cell lines were seeded in 96-well plates at 5000 and 15,000 cells/well, respectively, in 100 μL fresh cell medium. The day of the experiment, cells were added to 20 µL of each compound (in triplicate) and incubated at 37 °C for 96 h, unless otherwise specified. The MTT reagent (Sigma-Aldrich, 100 μL, 1 mg/mL in cell media) was added into each well and cells were incubated for 1.5 h at 37 °C. The medium was carefully removed and the blue precipitate was solubilised by adding 100 μL of DMSO. Cell viability was measured by quantifying absorbance at 570 nm using a 96-well plate reader (Flx-Xenius XM, Safas, Monaco). Viability ± SDs were calculated from three independent experiments, using the following equation:(5)$$\mathrm{Viability}=\frac{A-A_{0}}{A_{\mathrm{vehicle}}-A_{0}}\times 100$$
where A is the absorbance measured at 570 nm for the evaluated well, A_0_ is the mean of three wells where the medium was replaced with deionised water for the whole experiment (osmotic shock cell lysis), and A_vehicle_ is the mean of three wells treated with 20 µL of pure HBG buffer.

The in vitro cytotoxicity assays discussed in this work are provided in Figure 3A, Figure 4 and Appendix A.

### 2.4. In Vivo Experiments

Six-week-old female Balb/c nude mice were purchased from Janvier Laboratories (Le Genest Saint Isle, France). Animals were maintained under a controlled environment (20 ± 2 °C) with a relative humidity of 50 ± 10% and a 12 h/12 h light-dark cycle in ventilated GM500 cages (Techniplast, Milan, Italy). The bedding consisted of wood chips (Safe^®^ SF14, Safe laboratories, Augy, France) and cages were enriched with mouse house and nestlets pads. Food (autoclavable diet, Safe^®^ DO4, Safe laboratories, Augy, France), and tap water was available ad libitum. For heterotopic xenografts of SKBR3 cells, 5 × 10^6^ cancer cells were resuspended in 50% DMEM + 50% Matrigel^®^ HC (Corning, 354248) and were implanted subcutaneously into the flank of healthy 7-week-old female mice. Nine weeks after SKBR3 cell xenografts, mice were randomised based on tumour volume to obtain three homogenous groups then treated on day 0 (first treatment) and 9 via intravenous route with vehicle (HBG) or 1.8 nmol/kg of ATNP or cNP. Body weight monitoring is reported in Appendix A.

## 3. Results and Discussion

The two AOCs were synthesised using the previously described “plug and play” strategy. Briefly, in a first step, both antibodies were reacted with 4-azidobenzoyl fluoride to yield azido-antibodies (“plug” step). These were then conjugated to bicyclononyne-functionalised 37 mer single-stranded DNA (BCN-ssDNA) oligonucleotides through a strain-promoted azide-alkyne cycloaddition (SPAAC) reaction (“play” step). Thanks to the excellent bioorthogonality of SPAAC and quantitative conversion, the final degree of conjugation (DoC) is the same as the one obtained from the plug step [43].

While using a large number of equivalents of both 4-azidobenzoyl fluoride and the above BCN-ssDNA would lead to highly conjugated species that would tend to precipitate, the use of an insufficient number of equivalents would leave a significant portion of the antibodies unconjugated, and thus, not able to coat the cNP. The sweet spot between these two extremes was found when using three equivalents of 4-azidobenzoyl fluoride and performing the reaction in PBS 2X. The resulting conjugates were then purified using size-exclusion chromatography (SEC) to eliminate unconjugated BCN oligonucleotides and AOCs were obtained with an average DoC of 2.9. Based on native mass spectrometry (MS) and SDS PAGE analysis (Figure 2(Aa,Ab)), with mostly DoC = 2 and DoC = 3 species, and less than 5% unconjugated antibodies.

A polydiacetylenic (PDA) amphiphilic monomer bearing the tyrosine-histidine cationic head group was conveniently obtained in two steps: starting from our previously described tyrosine-PDA monomer [42], HBTU/HOBt-mediated coupling with a Boc-protected histidine was followed with the acidic deprotection of the Boc group in one pot (see SI for detailed protocol). Following a previously reported protocol [46], the obtained tyrosine-histidine PDA amphiphilic monomer was sonicated in water to allow its self-assembly into micelles, before being UV-irradiated at 254 nm to trigger photopolymerisation. The photopolymerised PDA micelles were analysed by dynamic light scattering (DLS) and were found to have a diameter of 7.3 nm and a ζ potential of +77.9 mV (see Figure 2(Ba,Bb), and SI for detailed protocol). As expected, the small size and highly cationic surface of these micelles are in accordance with what we observed for previously described histidine PDA micelles [34,46,47]. Based on the measured diameter of 7.3 nm for the PDA micelle (i.e., volume of about 203 nm^3^) and the estimated volume of about 0.92 nm^3^ for the monomer (using the formula described by Zhao et al. [45]), we calculated that each micelle was composed of an average of 220 monomers (see SI for detailed calculation), which is similar to the composition of previously reported micelles [48]. This number will serve as the basis to determine AOC/cNP ratios for all formulations.

The gel mobility shift assay was used to assess the capacity of the PDA micelle to complex unconjugated oligonucleotides (see Appendix A), and thus, the soundness of the supramolecular assembly strategy. A given quantity of fluorescein-labelled 37 mer ssDNA was incubated with various quantities of PDA. Satisfactorily total complexation (as evidenced by the absence of free ssDNA band on the gel; see Appendix A) was observed for ssDNA/cNP ratios of 1.2 and lower, indicating that a single particle could strongly complex up to one ssDNA molecule. The same experiment was performed with a fluorescent AOC and similarly showed that a single particle could not associate strongly with more than one AOC (see Figure 3C and Appendix A). Thus, the AOC/cNP ratio should not exceed 1 in order to avoid the presence of non-complexed species in solution.

To select the optimal AOC/cNP ratio in terms of biological activity and in vivo compatibility of Antibody-Toxic Nanoparticles (ATNPs), a series of ATNPs at AOC/cNP ratios lower than 1 (0.1, 0.2, 0.4, 0.7 and 1) was prepared. Two cell lines were used: SKBR-3 (HER2^+^), which is recognised by trastuzumab, and MDA-MB-231 (HER2^−^), which is not recognised by trastuzumab. We incubated these cells with either the non-complexed cNP or the five ATNPs at four concentrations of cNP (11 nM, 23 nM, 45 nM and 91 nM) for four days, in the presence of 10% of FBS, and measured the cells’ viability using the MTT assay (Figure 3A). The first interesting observation was that in all cases cytotoxicity towards the HER2^−^ MDA-MB-231 cell line was low (Figure 3A). Thus, it appears that the system by itself does not possess intrinsic nontargeted toxicity. In contrast, cytotoxicity towards the HER2^+^ SKBR-3 cell line was found to be very high for the ATNP complexes, in particular those with an AOC/cNP ratio ≥ 0.2 (Figure 3A). In addition, a dose response could be observed with increasing toxicity for increasing amounts of cNP. Of note, T-ssDNA conjugates did not by themselves reduce the cell viability of either cell lines in the absence of complexed cNP (see Appendix A). These encouraging results demonstrates that supramolecular cNP conjugation enables the triggering of selective cell killing.

Moving toward in vivo compatibility, it was required to validate the size and charge of the various supramolecular complexes. As the ratio of AOC/cNP increased from 0 to 1.0, DLS measurement showed that the sizes of the particles gradually increased from 7.3 nm to more than 200 nm. It has been shown that particles smaller than 10 nm are rapidly eliminated in vivo by renal clearance, while those larger than 100 to 200 nm tend to accumulate non-specifically in the liver and the spleen [21,49]. The optimal size for NPs intended for in vivo application is generally considered to fall within this 20–150 nm range. Based on these considerations, the AOC/cNP ratios of 0.7 and 1.0 were not further investigated because they led to ATNPs larger than 150 nm (Figure 3B).

Concerning the overall charge of ATNPs, when the AOC/cNP ratio was increased from 0 to 1.0, the ζ potential decreased from +76 to −7 mV. This trend accounts for the increasing neutralisation of the positive micelle’s charges by the negative charges brought by the oligonucleotide strand from AOCs. It has been shown that cationic particles interact non-specifically with anionic cellular membranes and circulating proteins, and tend to have shorter circulating half-lives than neutral or slightly negatively charged particles [21,50]. Consequently, ATNP with an AOC/cNP ratio of 0.4 seemed optimum for in vivo application since it showed both a close-to-neutral surface charge (+4 mV) and a small size (73 nm; see Figure 3).

At this optimum AOC/cNP ratio, further biological investigations were performed to further validate antibody-dependent cell line cytotoxicity. A mock, non-targeting ATNP was prepared from a PDA micelle and Rituximab-ssDNA conjugate (AOC/cNP ratio of 0.4). Both SKBR-3 (HER2^+^) and MDA-MB-231 (HER2^−^) cell lines were tested against Trastuzumab- and Rituximab-based ATNPs. Satisfactorily, while ATNP kept its high selectivity for HER2+ cells, mock ATNP showed a toxicity comparable to that of the free cNP (see Figure 4 and Appendix A). This antibody-dependent toxicity was further validated by a competition experiment in which pre-incubation with an excess of free, non-conjugated trastuzumab (1 µM) was shown to strongly protect SKBR-3 cells from the cytotoxicity of ATNP, without having any impact in the case of MDA-MB-231 (see Appendix A).

The robustness of supramolecular ATNP was further evaluated in even more stringent conditions. It is known that proteins contained in the blood are adsorbed onto the surface of nanoparticles, forming a “protein corona” which changes their behavior and hinder their active targeting [7,51,52]. While the previously mentioned cytotoxicity experiments were performed in a growth medium supplemented with 10% FBS, experiments were repeated in a 1:1 medium/FBS mixture that usually inhibits nontargeted cNP-Oligo based transfection. As shown in Figure 4, this change had little to no impact on the selectivity and cytotoxicity profile. Interestingly, under these new conditions, both the mock ATNP and the non-complexed cNP presented even less toxicity toward both cell lines. This is in stark contrast with more classic ADCs, whose payloads are highly cytotoxic on their own and strongly limit the maximum tolerated dose. In ATNP, it thus appears that each component of the system is non-toxic on its own, and yet, the complex they form is highly active, even under the most stringent conditions.

In order to gain deeper insight on the cNP/cell interaction kinetics and subsequent cell death, we repeated cytotoxicity assays, but this time incubated cells for only 4 h at either 37 or 4 °C before resuspending cells in fresh culture medium (10% FBS) for the following 92 h. Interestingly, under these conditions the toxicity profiles of the three constructs were very similar to those obtained with continuous incubation. This indicates that the amount of ATNP bound and further internalised to HER2^+^ cells in 4 h of contact is sufficient to exert very significant cell killing (80%) (See Appendix A). No systemic exposure seems to be needed.

Finally, to validate whether these positive in vitro results would translate in vivo, three groups of SKBR-3 xenograft-bearing Balb/c nude mice were intravenously injected two times with 45 pmol (1.8 nmol/kg) of either ATNPs, bare cNPs, or the vehicle (HEPES buffered Glucose, HBG) on days 0 and 9 (See Figure 5 and Appendix A).

ATNP 0.4 was chosen as the lead because it had characteristics suitable for in vivo experiments (i.e., small size and close to neutral overall charge) and showed strong in vitro activity. The non-treated control group (injected with HBG buffer, or vehicle) and the cationic nanoparticle alone were chosen as controls. The outcome would inform on the behavior of free cytotoxic agents that might be released in the bloodstream upon premature deconjugation and, by comparison, highlight the effect of antibody-mediated active targeting.

Evolution of the tumour sizes in the ATNP and cNP groups was compared to that in the vehicle group. In the HBG group, the tumour grew slightly during the first 12 days after the first compound administration (day 0) and remained between 120–130% of their initial volume for the last 16 days. In the cNP group, tumours grew similarly to in the HBG group and were stable for the last 16 days. In the ATNP group, however, a statistically significant tumour size reduction was observed after seven days relative to both the cNP and vehicle groups. Over time, the ATNP group showed steady tumour size reduction with no apparent toxicity despite two injections of the compound (see Appendix A).

This experiment underlines different interesting features. The free cNP bearing tyrosine-histidine dipeptide and cationic polar head group that we developed for this application seem not to show any toxicity on their own. This is in sharp contrast with classical payloads used in the ADC strategy and supports our original hypothesis. Once assembled via supramolecular interactions with tumour-targeting Ab and injected, stable regression could be observed without any weight loss. In accordance with in vitro experiments performed at high serum content, sufficient amounts of ATNP reached the tumour in order to trigger stable regression and again underlined the good safety profile of ATNP. While further studies will be necessary to completely understand the fate of ATNP in vivo and identify possible bottlenecks, these proof-of-concept results clearly demonstrate the potency of this novel approach combining cNP as toxic payload and supramolecular conjugation as a construction principle.

## 4. Conclusions

Supramolecular interaction of polycationic diacetylene micelles and polyanionic antibody-oligonucleotide conjugates formed complexes with easy-to-tune biophysical and biological properties. This versatile assembly strategy allowed for optimization of the formulation conditions in order to afford small, neutral, highly active and selective Antibody-Toxic Nanoparticle conjugates. Cell-specific delivery of cationic nanoparticles to HER2^+^ cells through antibody-antigen specific binding led to cell death at as low as 40 nM of toxic cNP despite the presence of 50% FBS. Interestingly, and in contrast with small molecule drug payloads that target precise biological processes and often require long ADC exposure, ATNPs showed consistent antigen-binding mediated toxicity after just 4 h of exposure. In addition, we showed that the free cNPs developed for this project did not trigger unwanted side toxicity on their own. This is sharp contrast to classical small molecule payloads that, upon premature deconjugation in the blood stream, trigger important side toxicity. In vivo experiments showed robust tumour regression in a xenografted mice in vivo model, thus confirming that enough cNP had reached the tumour, while no signs of side toxicity could be seen even after two injections of ATNP or cNP. The present work opens prospects for the further development of cationic nanoparticles as a novel type of cytotoxic payload in antibody-based targeted drug delivery systems. Bio-inspired or bioactive NP used to enlarge the therapeutic scope of ATNPs can be envisioned, such as the addition of encapsulated hydrophobic compounds, or the addition of an intracellular targeting element [53,54].

## Figures and Tables

**Figure 1 pharmaceutics-15-01643-f001:**
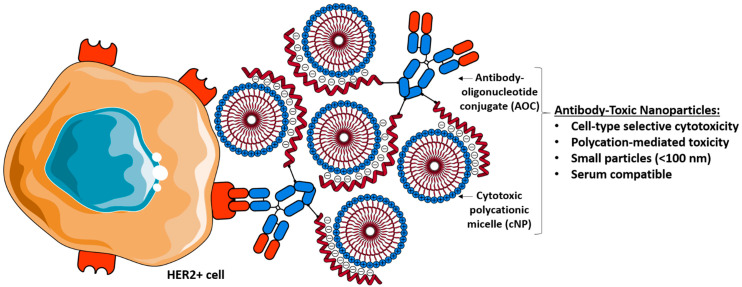
General scheme of the ATNP system: non-covalent complexation of cytotoxic polycationic micelles with antibody–oligonucleotide conjugates into small nanoparticles, allowing for the selective killing of cancer cells. This representation of the ATNP complex is consistent with an AOC/cNP ratio of 0.4 optimised for in vivo application (see below).

**Figure 2 pharmaceutics-15-01643-f002:**
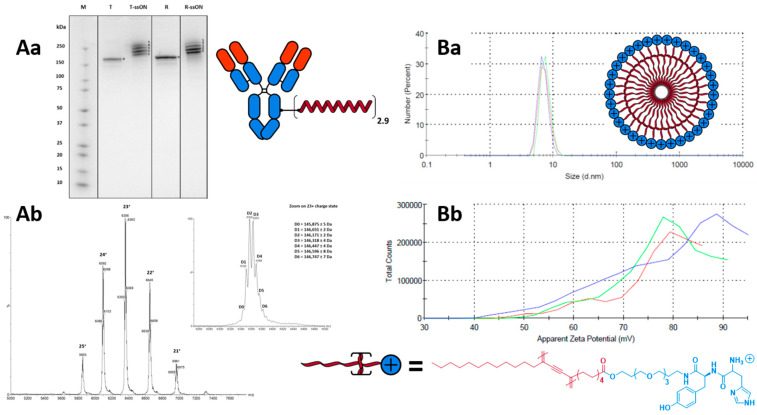
Characterisation data for the targeting antibody-oligonucleotide conjugate (**A**) and the cytotoxic cationic micelle payload (**B**) that constitute the targeted system. Antibodies are first functionalised with azido groups, allowing for the installation of BCN-functionalised oligonucleotides through a strain-promoted alkyne-azide click reaction. SDS-PAGE (4–15%) of the final conjugate (**Aa**) and native MS of the azido-functionalised intermediate (**Ab**) indicate degrees of conjugation of 2.9 and 2.8, respectively. Photopolymerised polydiacetylenic micelles featuring a cationic tyrosine-histidine dipeptide head group were analysed via DLS, presenting a size of 7.3 nm (**Ba**) and ζ potential of +77.9 mV (**Bb**).

**Figure 3 pharmaceutics-15-01643-f003:**
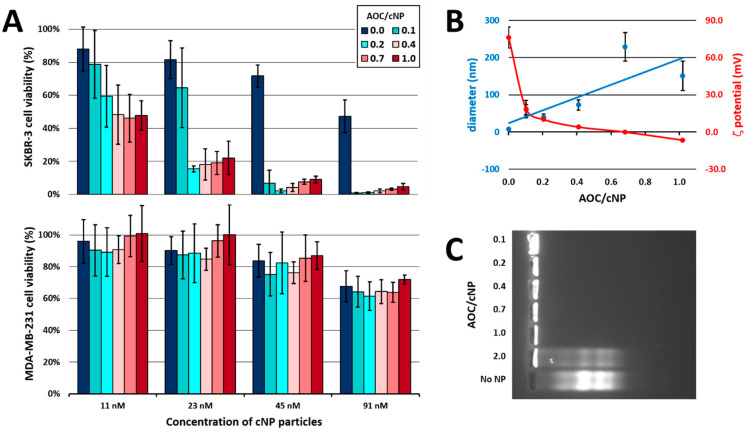
Micelles were complexed with T-ssDNA conjugate at molar ratios of 0 (micelle alone), 0.1, 0.2, 0.4, 0.7, and 1 conjugate per particle. For each complex, we evaluated: cytotoxicity after 96 h of incubation in 10% FBS containing culture medium towards both the targeted, HER2^+^, SKBR-3 (**A**) top, and the non-targeted, HER2^−^, MDA-MB-231 (**A**) bottom, cell lines, using the MTT assay (error bars represent the standard deviation from at least three assays); and the size (blue line) and ζ potential (red line) using dynamic light scattering (**B**) (error bars represent the standard deviation from three measurements). Using fluorescein-labelled T-ssDNA conjugates, we prepared complexes at molar ratios of 0.1, 0.2, 0.4, 0.7, 1 and 2 conjugates per particle, and analyzed them using agarose gel electrophoresis (1.3% *w*/*v*, 80 mV, 60 min, tris-acetate running buffer) alone with the non-complexed conjugate (**C**).

**Figure 4 pharmaceutics-15-01643-f004:**
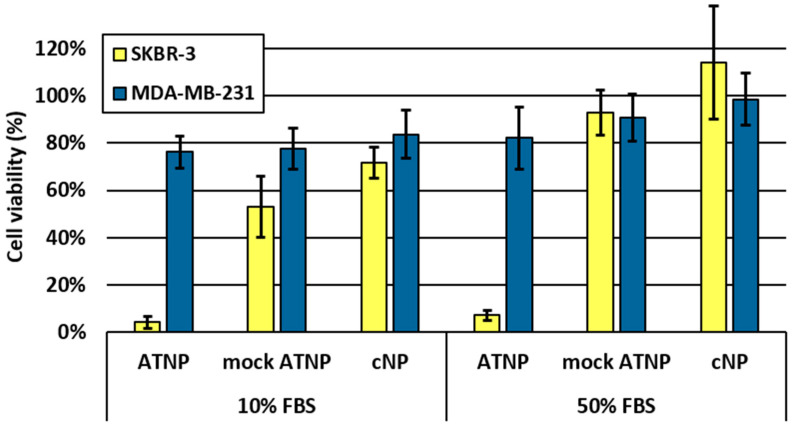
Cellular viability of both the HER2^+^ SKBR-3 and the HER2^-^ MDA-MB-231 cell lines after 96 h of continuous incubation at 37 °C in either 10% or 50% FBS supplemented culture medium, and in the presence of 45 nM (particles concentration) of either the ATNP (T-ssDNA/micelle ATNP complex), the mock ATNP (R-ssDNA/micelle complex) or the cNP (cationic micelle), as measured using the MTT assay. For the ATNP and mock ATNP, the complexes were prepared at a molar ratio of 0.4 AOC/cNP. Viability is expressed as a percentage relative to vehicle-treated (i.e., 20 µL of pure HBG buffer), and the error bars represent the standard deviation calculated from at least three independent assays.

**Figure 5 pharmaceutics-15-01643-f005:**
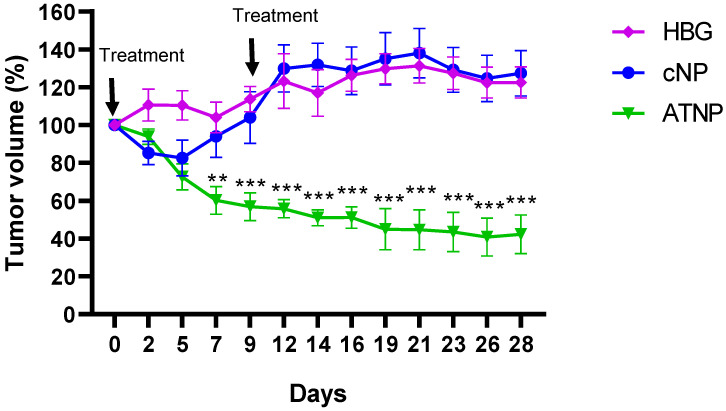
In vivo activity of ATNPs in a mouse model with SKBR3 heterotopic xenograft. Balb/c nude mice with SKB3 tumours were treated intravenously with 1.8 nmol/kg of ATNP and cNP or vehicle (HBG) on day 0 and 9. The ordinate axis reports tumour volume (%), dots representing means and error bars are S.E. values (n = 4 to 6/group). Statistical analysis consisted of two-way ANOVA followed by Tukey’s multiple comparisons test and was conducted with Prism software (** *p* ≤ 0.01; *** *p* ≤ 0.001 vs. cNP and HBG group).

## Data Availability

The data presented in this study are available in the article or Appendix A.

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
