# Peer review of "Targeted Anticancer Agent with Original Mode of Action Prepared by Supramolecular Assembly of Antibody Oligonucleotide Conjugates and Cationic Nanoparticles"

_pharmaceutics, 2023, doi:10.3390/pharmaceutics15061643_

Round 1

Reviewer 1 Report

The manuscript “Targeted anticancer agent with original mode of action prepared by supramolecular assembly of antibody oligonucleotide conjugates and cationic nanoparticles” describes the study on the non-covalent complexation approach and presents some data on the in vivo stability of the resulting structure. This is a very interesting and important concept, which is still under discussion in the scientific community due to often conflicting data. The results would be of interest to readers of the Journal. Unfortunately, the manuscript is poorly organized, some mandatory sections are missing, and the proofreading is poorly conducted. This makes the reading somewhat difficult and obscures the technical content. The in vivo data may also need additional discussions. Please see comments below.

- The manuscript seems to present data in the chronological order (“We first used…”, “We then compared, “we repeated our cytotoxicity assays” , “we then sought…”, “we decided to repeat those experiments”) and is very descriptive in some sections explaining how the selection of drug candidates was performed. There are also a lot of values listed in the text (specific compositions, zeta potentials, wavelengths used for polymerization, concentrations of NPs, etc., etc.) This makes the manuscript incredibly difficult to read. Are those numbers really critical for understanding the content and can they be moved to SI? How important some of them for understanding the content and conclusions? The use of chronological approach to data presentation along with huge number of details makes it difficult to follow the research strategy.

- The main results of in vivo study, which are presented in Figure 5, are described extremely briefly in a single short paragraph at the end of page 7. This is surprising given how much attention is given to the intermediate details, as mentioned above. In fact, even one of three formulations HBG - vehicle is not described. Most importantly, the Reviewer wonders if all appropriate controls were considered to support the conclusions? For example, why ‘mock ATNP’ wasn’t used as a control? Most importantly, to make the conclusion on the biological activity of the NON-COVALENT conjugate - the main conclusion of the paper, it is very important to make the injection of two formulations components separately. If the authors decided not to include this in the study, it is important to discuss the reasons for it. Given the overall skepticism of the scientific community and conflicting published data on the stability on non-covalently linked components, such discussion would be critical. Finally, given a series of different ATNPs described in the paper, why not to list specifics of the in vivo tested formulation here.

- Given the focus of the manuscript on non-covalent complexation (conjugation) concept, it would be useful to have a very brief discussions on some previously published reports.

- The manuscript lacks Materials and Methods section. Form some reason, the whole section was placed in the Supplementary Information. This section (2. Materials and Methods) is obligatory. Please see authors instructions and Journal’s template for more information.

- Some Figures are not listed in numerical order. For example, Figure 4 is first referred to on Page 5, whereas Figure 5 is already mentioned on page 4.

- The manuscript contains in vivo data. However, there is no “Institutional Review Board Statement”. Please see authors instructions and Journal’s template for more information.

- Multiple error messages are in the text. (ATNP; Error! Reference source not found.)

- The corresponding author is not clearly shown.

- The abstract is too long (286 words) and not well structured. It also discusses prospects for new research for which no data presented in the manuscript. Please see Journal’s ‘Instructions for Authors’ for guidance: “The abstract should be a total of about 200 words maximum. The abstract should be a single paragraph and should follow the style of structured abstract…The abstract should be an objective representation of the article: it must not contain results which are not presented and substantiated in the main text and should not exaggerate the main conclusions.”

- Figure 1 is difficult to understand without clear labeling of the components .

- Some methods are not adequately described. For example, authors use the term ‘gel retardation’ assay and refer to Figure S5, which does not refer to that term.

- No author contributions, funding, conflict of interest, data availability, etc. sections.

-  Are all statements in the conclusion are supported by the data? For example: “ATNPs working by biophysical destabilization of endolysosomes”? Also, conclusions list multiple poorly defined terms: “with tenable properties”, “classical payloads”, hard-to-handle side-toxicity, secondary effects, fast activity of the system,

Author Response

The reply to reviewer are detailed is the attachement

Reviewer 2 Report

This manuscript deals with "Targeted anticancer agent with original mode of action prepared by supramolecular assembly of antibody oligonucleotide conjugates and cationic nanoparticles." I suggest a minor correction and require a detailed clarification. A correction should be addressed by the authors as follows: The abstract is not well organized; the sentences are incomplete, and there is no sense of continuity. It would be feasible if you included the significance of the current study in the abstract. A brief description of how the authors selected information from the literature in the databases, as well as what time period they searched for, is missing. The authors should justify and expand the information on the advantages of cationic nanoparticles for biomedical applications. Authors should specify the main experimental conditions used based on the evidence from the literature. Where they briefly describe the most important data reported in the literature in a homogeneous manner and reinforce the relevance of cationic nanoparticles as novel alternatives. Authors should discuss whether the use of cationic nanoparticles  represents a solid alternative to existing therapeutics. Also, please discuss the use of method using green nanomaterials to targeting cells and mitochondria . Please add the below studies to your manuscript in the discussion section and bold your study novelties:

-Ramazanli, V. N., & Ahmadov, I. S. (2022). SYNTHESIS OF SILVER NANOPARTICLES BY USING EXTRACT OF OLIVE LEAVES. Advances in Biology & Earth Sciences Vol.7, No.3, 2022, pp.238-244 -

-Baran, A., Fırat Baran, M., Keskin, C., HatipoÄŸlu, A., Yavuz, Ö., İrtegün Kandemir, S., ... & Eftekhari, A. (2022). Investigation of antimicrobial and cytotoxic properties and specification of silver nanoparticles (AgNPs) derived from Cicer arietinum L. green leaf extract. Frontiers in Bioengineering and Biotechnology, 10, 263.

Author Response

The reply to reviewer 1 and 2 are detailed is the attachement

Round 2

Reviewer 1 Report

The manuscript can now be accepted.